# Duodenojejunal Omega Switch Surgery Reduces Oxidative Stress Induced by Cafeteria Diet in Sprague-Dawley Rats

**DOI:** 10.3390/nu14194097

**Published:** 2022-10-02

**Authors:** Jakub Poloczek, Wojciech Kazura, Elżbieta Chełmecka, Katarzyna Michalczyk, Jerzy Jochem, Janusz Gumprecht, Dominika Stygar

**Affiliations:** 1Department of Rehabilitation, 3rd Specialist Hospital in Rybnik, Energetyków 46 Street, 44-200 Rybnik, Poland; 2Department of Internal Medicine, Diabetology and Nephrology, Faculty of Medical Sciences in Zabrze, Medical University of Silesia, Poniatowskiego 15 Street, 40-055 Katowice, Poland; 3Department of Physiology, Faculty of Medical Sciences in Zabrze, Medical University of Silesia, Jordana Street 19, 40-055 Katowice, Poland; 4Department of Statistics, Department of Instrumental Analysis, Faculty of Pharmaceutical Sciences in Sosnowiec, Medical University of Silesia, Poniatowskiego 15 Street, 40-055 Katowice, Poland

**Keywords:** antioxidants, bariatric surgery, cafeteria diet, duodenojejunal omega switch (DJOS), high-fat diet, high-sugar diet, lipid peroxidation, obesity, oxidative stress

## Abstract

Over-nutrition with cafeteria diet leads to glycemic control failure and subsequent obesity. Bariatric surgery remains the ultimate treatment option, and when complemented with specific dietary protocol, it may mitigate the effects of oxidative stress induced by a cafeteria diet. The study measured antioxidant marker activity: superoxidase dismutase (SOD) and ceruloplasmin (CER), total antioxidant capacity (TAC), and lipid peroxidation marker concentrations: lipofuscin (LS) and malondialdehyde (MDA), in the plasma of 56 Sprague-Dawley rats fed with a cafeteria (HFS) or a control (CD) diet and subjected to duodenojejunal omega switch (DJOS) or control (SHAM) surgery. The diet change after the surgery (CD/HFS or HFS/CD) strongly influenced SOD activity in DJOS- and SHAM-operated rats, but SOD activity was always higher in SHAM-operated rats. Every dietary protocol used in the study increased CER activity, except for the CD/CD combination. Cafeteria diet consumed before or after either of surgeries led to decrease in TAC levels. DJOS and no change in diet reduced MDA levels. DJOS reduced LS levels, but its beneficial effect was deteriorated by selected dietary protocols. The cafeteria diet negatively affected the positive impact of DJOS surgery, but SOD, CER, MDA, and LS were significantly lower in rats that underwent DJOS, suggesting that eight weeks of dietary treatment before and after the surgery did not totally dilapidate the effects of the bariatric treatment.

## 1. Introduction

The cafeteria diet used in animal studies enables the reproduction of the effects of the highly palatable and highly caloric Western diet [1,2]. The Western diet is mainly composed of foods rich in refined sugars and saturated fats, such as processed meats, sweets, non-alcoholic drinks, snacks, and condensed milk, added to a standard diet [3]. In experimental animals, the cafeteria diet efficiently induces hyperinsulinemia, hyperglycemia, glucose intolerance, increased adiposity, and hepatosteatosis—disorders similar to those responsible for human metabolic syndrome [4] and also greater food consumption and increased weight gain [2,5], leading to a phenotype of exaggerated obesity. Hence, the cafeteria diet is suitable for modeling the metabolic disorders of human obesity in animals [4].

Human obesity results not only from excessive saturated fat intake but also from various nutritional and lifestyle factors linked to the overconsumption of highly processed, low-quality foods [6]. Together with stress and chronobiological rhythm disruption, also characteristic of modern Western lifestyle, there is a further contribution to the increase in metabolic syndrome prevalence [4]. On the physiological level, obesity results in oxidative stress and chronic low-grade inflammation [7,8,9] that is controlled by the various factors keeping equilibrium between pro- and antioxidative processes [10,11,12]. Oxidative stress was observed in experimental diabetes as early as 1982 [13], and it has been found to play an essential role in all cases of diabetes mellitus (particularly type 2 diabetes mellitus, T2DM) and the pathogenesis of diabetic complications. T2DM is associated with increased oxidative stress resulting from several abnormalities, including hyperglycemia, inflammation, and dyslipidemia [6,7]. In turn, high levels of reactive oxygen species (ROS) can act as secondary messengers and regulate the biological function of various proteins. The dynamic modification of intracellular redox sensors by ROS is defined as redox modification and is similar to other posttranslational modifications such as protein phosphorylation, acetylation, or ubiquitination, which play an important role in diabetes development [10]. 

Obesity treatment aims at permanent weight loss, achieved by nutritional therapy, as well as metabolic disorders treatment. However, nutritional therapy often fails in morbidly obese patients, and for such patients, bariatric surgery remains the only reliable option [14] to ensure permanent weight loss. Additionally, bariatric surgery induces positive changes in glucose tolerance, thus providing diabetes control [15,16], as well as in signaling pathways and concentrations of different proteins [10,16].

We hypothesized that bariatric surgery complemented with a certain dietary protocol would mitigate the effects of oxidative stress induced by the cafeteria diet. This study aimed to measure the activity of the antioxidant markers superoxidase dismutase (SOD) and ceruloplasmin (CER), the levels of total antioxidant capacity (TAC), and the concentrations of lipid peroxidation markers lipofuscin (LS) and malondialdehyde (MDA) in the plasma of Sprague-Dawley rats fed a cafeteria diet and subjected to bariatric surgery, namely, duodenojejunal omega switch (DJOS).

In this article, for the first time, we analyzed the effects of a high-fat/high-sugar diet on the selected oxidative stress markers in relation to bariatric surgery.

## 2. Materials and Methods

The experimental design and procedures were approved by the Ethical Committee for Animal Experimentation of the Medical University of Silesia (Katowice, Poland) (58/2014). The institutional and national guidelines for animal care and use (Directive 2010/63/EU) were followed. The number of rats involved in the procedure was minimized according to the 3Rs guidelines for the humane treatment of animals [17]. The survival rate of animals included in the study was 100%.

### 2.1. Study Subject

Fifty-six male 7-week-old Sprague-Dawley rats (Charles River Breeding Laboratories, Wilmington, MA, USA) weighing 250–275 g were included in the study. Rats were kept in a plastic cage and fully controlled conditions: 12/12 h of dark/light cycle, at 23 °C, with unrestricted access to water and food.

### 2.2. Study Design

After 7 days of acclimation, the animals were randomly assigned to the experimental groups. In the first part of the experiment, the rats were maintained for eight weeks on experimental diets: control diet (CD) (n = 28) or high-fat/high-sugar diet (HFS) (n = 28), inducing obesity. The detailed characteristics of CD and HFS diets are provided in Table 1.

After this time, rats from each experimental group (CD and HFS) were randomly assigned to two subgroups that each underwent a different type of surgery: control (SHAM) (n = 14) or duodenojejunal omega switch (DJOS) (n = 14), resulting in four subgroups in total: CD/SHAM, CD/DJOS, HFS/SHAM, and HFS/DJOS. The second part of the experiment (after the surgery) assumed a maintenance of half of the rats from each subgroup on the same diet as before the surgery (n = 7) (CD/SHAM/CD, CD/DJOS/CD, HFS/SHAM/HFS, HFS/DJOS/HFS), and the other half of the rats from each subgroup on a different diet to that before the surgery (n = 7) (CD/SHAM/HFS, CD/DJOS/HFS, HFS/SHAM/CD, HFS/DJOS/CD) for a further eight weeks (Figure 1).

Eight weeks after DJOS and SHAM surgery, the antioxidant markers such as total superoxide dismutase (SOD) activity, ceruloplasmin (CER) concentration, and total antioxidant capacity (TAC), and lipid peroxidation markers, such as lipofuscin (LF) and malondialdehyde (MDA) concentration, in the rat plasma were analyzed.

### 2.3. Experimental Procedures

#### 2.3.1. Control (SHAM) and Duodenojejunal Omega Switch (DJOS) Surgery

Control (SHAM) and duodenojejunal omega switch (DJOS) surgeries were performed as described in the work of Stygar et al. [18].

Oxygen flow of 2 L/min and 2% isoflurane (AbbVie, Ludwigshafen, Germany) under spontaneous breathing was used to maintain the anesthesia. Xylazine (5 mg/kg, ip; Xylapan, Vetoquinol Biovet, Puławy, Poland) was used to achieve analgesia. Gentamicin (gentamycin 40 mg/mL, Krka, Warszawa, Poland) was used as the antibiotic prophylaxis.

A 3–4 cm midline incision was made to access the abdomen. The duodenum was separated from the stomach slightly distally to the pyloric sphincter. The proximal part of the small bowel and the duodenum was excluded from the bowel content passage and closed using PDS 6/0 (Ethicon, Cincinnati, OH, USA). The end-to-side anastomosis (duodeno-enterostomy) was positioned at one-third of the small intestine total length to restore the bowel content passage. The mesentery was closed with PDS 6/0.

During SHAM surgery, the gastric tract was cut at the site analogous for duodenum separation during DJOS surgery, and it was immediately reattached to the stomach at the same position, and thus the intestinal food passage was restored.

Carprofen (4 mg/kg, sc; Rimadyl, Pfizer, Zürich, Switzerland) was used to achieve analgesia in the post-operative period for three consecutive days.

#### 2.3.2. Blood Collection and Plasma Separation

Eight weeks after the surgery, the right tail vein was cannulated with the 26-gauge cannula, and 700 µL of blood was collected to tubes containing 10 μL (7.2 mg) of EDTA (Sigma-Aldrich, Burlington, MA, USA). Next, the blood was centrifuged at 4000 rpm for 10 min at 4 °C. The separated plasma was collected, snap-frozen in liquid nitrogen, and stored at −80 °C until further analysis. Protein concentration in the plasma was determined using the Lowry method with bovine plasma albumin as the standard [19].

#### 2.3.3. Antioxidant and Lipid Peroxidation Markers Analysis

##### Superoxide Dismutase (SOD) (EC 1.15.1.1) Activity

SOD activity was determined with the Oyanagui method [20]. SOD activity was expressed as nitrite units (NU) per milligram of protein in plasma, where 1 NU equals 50% inhibition of nitrite ion formation under the method’s condition.

##### Ceruloplasmin (CER) (EC 1.16.3.1) Activity

Ceruloplasmin (CER) activity was assessed according to the Richterich method [21], where ceruloplasmin oxidizes the translucent p-phenyl diamine to a blue-violet dye. In this method, the absorbance of the tested and control sample is read at 560 nm. The results are expressed in mg/mL.

##### Total Antioxidant Capacity (TAC) Level

TAC was measured using the Erel method [22]. In this method, a translucent reduced ABTS molecule (2,2′-azino-bis(3-ethylbenzothiazoline-6-sulfonate) is oxidized to blue-green ABTS^+^. Mixing ABTS^+^ with any substance that can be oxidized reverses ABTS^+^ to its original translucent form. TAC is expressed in mmol/L. 

##### Malondialdehyde (MDA) Concentration

Malondialdehyde (MDA) concentration was measured using the method of Ohkawa et al. [23], using the reaction with thiobarbituric acid. MDA concentration in the plasma was calculated against the standard curve prepared from 1,1,3,3-tetraethoxypropane and was expressed in µmol/L.

##### Lipofuscin (LPS) Concentration

Lipofuscin (LPS) concentration was measured as described by Tsuchida et al. [24]. The plasma was mixed with ethanol-ether (3:1 *v*/*v*), shaken, and centrifuged. The fluorescence intensity was determined at 345 and 430 nm for absorbance and emission, respectively. The results were expressed in relative lipid extract fluorescence (RF), where 100 RF corresponds to the fluorescence of 0.1 μg/mL quinidine sulfate in 0.1 N sulfuric acid.

### 2.4. Statistical Analysis

Statistical analysis was performed using a Statistica 13.0 data analysis software system (TIBCO Software Inc., Palo Alto, CA, USA). The normality of the distributions was verified using the Shapiro–Wilk test and quantile–quantile plots. The results for data with non-normal distribution were presented as median (lower-upper quartile), Me (Q_1_-Q_3_). The non-parametric Kruskal–Wallis test was used for comparisons of antioxidant and lipid peroxidation markers for each of the treatments depending on the applied dietary protocol. For the reader’s convenience, the figures show the combined results for both procedures. All the tests used were two-tailed. The statistical significance was set at a *p* < 0.05.

## 3. Results

In the case of rats from DJOS and SHAM groups with their diet changed after the surgery, we found statistically significant differences only for lipofuscin (LS) plasma concentration. Lower LS concentrations were noted in the HFS/CD groups as compared to the CD/HFS groups (*p*_DJOS_ < 0.05, *p*_SHAM_ < 0.05). The analysis of the remaining parameters showed no statistically significant differences between the groups with diet changed after the surgery in DJOS and SHAM animals (Appendix A).

In the case of the groups with an unchanged diet after the surgery, CD/CD and HFS/HFS, we found statistically significantly higher LS concentration in the plasma of rats of the HFS/HFS group, in both those subjected to the DJOS procedure and those subjected to the SHAM procedure (*p* < 0.05 for DJOS and SHAM). There was no difference in SOD activity and MDA concentration in both DJOS- and SHAM-operated rats. Moreover, we found no differences in CER activity and TAC levels in rats subjected to DJOS surgery. In the case of rats subjected to the SHAM surgery, we noted higher CER activity in the HFS/HFS group (*p* < 0.001) and a higher TAC level in the CD/CD group (*p* < 0.001) (Appendix A).

We found statistically significant differences in SOD activity depending on the dietary protocol, both in SHAM- (*p* < 0.001) and DJOS-operated rats (*p* < 0.001). When analyzing the dietary protocols individually, we observed higher SOD activity in SHAM-operated rats compared to DJOS-operated rats (Figure 2). In the case of rats subjected to SHAM surgery, we noted the lowest SOD activity in the CD/HFS and HFS/CD groups, and these results did not differ statistically from each other (*p* = 1). On the other hand, in the CD/CD and HFS/HFS groups, the SOD activity was significantly higher, and, as in the case of the groups with a changed dietary protocol, no statistically significant differences were found between them (*p* = 0.849). In the case of DJOS-operated rats, the groups with changed dietary protocol did not differ regarding SOD activity (*p* = 0.724). In contrast, lower SOD activity was observed in the HFS/HFS group in comparison with the CD/CD group (*p* < 0.001). 

We found statistically significant differences in CER activity depending on the dietary protocol, both after SHAM (*p* < 0.001) and DJOS surgery (*p* < 0.001). The lowest CER activity was observed in the CD/CD groups. Rats subjected to SHAM surgery showed higher CER activity than rats subjected to DJOS (Figure 3).

We found statistically significant differences in TAC levels depending on the dietary protocol, after both SHAM (*p* < 0.001) and DJOS surgery (*p* < 0.001). We noted the highest TAC levels in the CD/CD groups. Rats subjected to SHAM surgery presented lower TAC levels in all dietary groups compared to rats subjected to DJOS (Figure 4).

We found statistically significant differences in MDA concentration depending on the dietary protocol, both in SHAM- (*p* < 0.001) and DJOS-operated rats (*p* < 0.001). Individual analysis of the dietary groups showed higher LS levels in SHAM-operated rats than in DJOS-operated rats, except for the HFS/HFS groups, where similar levels of MDA were recorded (Figure 5). In the case of DJOS- and SHAM-operated rats, the lowest MDA concentrations were observed in the CD/CD and HFS/HFS groups, and no statistically significant differences were found between these diets (*p*_DJOS_ = 0.849, *p*_SHAM_ = 0.849, respectively). Higher MDA concentrations were noted in the rats from groups with a change in the dietary protocol. No statistically significant differences were found between the rats of the CD/HFS and HFS/CD groups (*p*_DJOS_ = 0.849, *p*_SHAM_ = 0.849).

We found statistically significant differences in LS concentration depending on the dietary regimen used, both in SHAM- (*p* < 0.001) and DJOS-operated rats (*p* < 0.001). We noted higher CER activity in SHAM-operated rats than in DJOS-operated rats for each feeding group (Figure 5). The lowest LS concentrations were observed in the CD/CD groups, and they differed statistically from those noted for the CD/HFS (*p*_DJOS_ < 0.001, *p*_SHAM_ < 0.001) and HFS/HFS groups (*p*_DJOS_ < 0.05, *p*_SHAM_ < 0.05). In the case of groups with a changed dietary protocol, the LS lower concentration was noted in the HFS/CD groups (*p*_DJOS_ < 0.05, *p*_SHAM_ < 0.05). The LF concentration in the plasma of rats from groups with changed dietary protocol and the HFS/HFS group was the same in SHAM- and DJOS-operated rats (Figure 6).

## 4. Discussion

In this study, we analyzed the combined effect of cafeteria diet and bariatric surgery (DJOS) on the levels of selected antioxidants and lipid peroxidation markers. We can make the following statements: (a) The change of diet after the surgery strongly influenced SOD activity in DJOS- and SHAM-operated animals. Regardless of the applied diet, SOD activity was higher in SHAM-operated than in DJOS-operated rats. The pattern of SOD activity was different for SHAM- and DJOS-operated groups. (b) Each dietary protocol used in the experiment increased CER activity, except for the CD/CD combination. (c) Cafeteria diet consumed before or after either of surgeries led to significant decrease in TAC levels both in DJOS- and SHAM-operated groups. (d) Bariatric surgery (DJOS) and the same diet consumed before and after the surgery (the CD/CD and HFS/HFS groups) were the factors significantly reducing the MDA level in the plasma. (e) Bariatric surgery reduced LS level, but its beneficial effect was deteriorated by selected dietary protocols. 

Chronic over-nutrition from a cafeteria diet leads to glycemic control failure resulting from insulin resistance and metabolic malformations. Oxidative stress and inflammation initiate and mediate pathological changes in the metabolic mechanisms [25]. The liver accumulates lipids and is infiltrated by immune cells. Muscles and other peripheral tissues become resistant to the physiological glucose stimulation [26]. 

Oxidative stress is considered the most important cause of complications developed in rats with diet-induced obesity. Oxidative stress in rats fed a high-fat diet also increases their glucose intolerance and insulin resistance [27,28]. Our previous studies in the same animal model reported that cafeteria diet impairs glucose metabolism and induces obesity, oxidative stress, and liver fat accumulation [11,29]. In the present study, we observed that the cafeteria diet induced significant disturbances in the antioxidant and lipid peroxidation markers in obese rats subjected to bariatric surgery. 

Superoxide dismutases are oxidoreductases comprising major cellular defense against oxidative stress that catalyze the dismutation of O_2_•− into H_2_O_2_, further degraded into oxygen and water by catalase (CAT). We observed that SOD activity was lower in the plasma of DJOS-operated rats, compared to SHAM-operated rats, regardless of the dietary protocols. It seems that DJOS surgery reduced the deleterious effect of cafeteria diet (HFS). The results are similar to those obtained by Ulla et al. They also observed significantly lower SOD activity in the liver of rats fed a high-energy diet and supplemented with *Syzygium cumini* seed powder, considered to have antioxidant and anti-inflammatory properties [30]. Ulla et al. showed that hepatic antioxidant activities of SOD, CAT, glutathione peroxidase (GPx), and glutathione (GSH) contents were significantly decreased in cafeteria-diet-fed rats as compared to normal diet rats [30]. Skrzep-Poloczek et al. [31] showed increased body mass and increased levels of oxidative stress markers, including total SOD activity in the erythrocytes and heart muscle of rats maintained on different dietary patterns, including high-fat diet and normal diet, from the SHAM group. However, they also observed the decrease in SOD activity in rats fed a high-fat diet after DJOS surgery [31].

Ceruloplasmin (a2-plasma glycoprotein) is an antioxidant protein transporting 95% of the Cu in blood [32,33]. Until recently, it was believed that ceruloplasmin is produced mainly in the liver. However, newer data suggest that it may also be of adipose tissue origin, as CER secretion and its circulatory levels were substantially higher in obese than in non-obese patients [34].

Kennedy et al. [35] found higher plasma Cu levels and CER activities in obese mice. Most of the changes in plasma Cu concentrations are associated with changes in the cuproenzyme ceruloplasmin [36]. Gletsu et al. [37] presented the long-term impact of *Roux-en-Y* gastric bypass surgery (RYGB) by measuring systemic copper concentrations before and during the 2-year period after the surgery [37]. They showed a steady decrease in CER activity within that time period, suggesting that copper status measured by ceruloplasmin worsens after RYGB [37]. In the presented study, CER plasma levels were lower in DJOS-operated rats compared to SHAM-operated rats, regardless of the dietary protocol. Such a result suggests that DJOS surgery might influence rats’ systemic Cu levels and indirectly CER activity in the plasma, as suggested by Gletsu et al. [37]. However, the cafeteria diet (HFS) increased CER levels both in DJOS- and SHAM-operated rats, with the highest levels noted for the HFS/HFS group of SHAM-operated rats. The observation would agree with reports by Arner et al., showing that CER is a novel adipokine [34]. Nevertheless, this hypothesis requires further research.

Total antioxidant capacity refers to the total capacity of the material, herein, the plasma, to neutralize the prooxidants generating reactive oxygen species. The TAC level reflects the synergistic or antagonistic effects of the various antioxidants present in the tested material [33,38,39]. In the presented study, TAC levels were the highest in the rats maintained on a control diet before and after surgery, regardless of its type, DJOS or SHAM. Every instance of introducing the cafeteria diet decreased TAC levels both in DJOS- and SHAM-operated rats. Nevertheless, TAC levels were higher in DJOS-operated rats than SHAM-operated ones. The observation suggests that the increase in antioxidative capacity was triggered by duodenojejunal omega switch surgery. 

The same results were observed for MDA levels, which were significantly lower in DJOS-operated rats than in SHAM-operated ones. MDA is an end product of non-enzymatic oxidative degradation of PUFAs. MDA level is quantitatively related to lipid per-oxidation, and that is why it is used a lipid peroxidation marker [40]. In this study, MDA levels were the highest in the SHAM-operated rats fed according to the changed dietary protocol: HFS/CD or CD/HFS. SHAM-operated animals fed with the same diet before and after the surgery, CD/CD or HFS/HFS, showed lower levels of lipid peroxidation. Levels of antioxidants and lipid peroxidation markers presented here depended more on the type of dietary pattern used pre- and post-operatively than on the bariatric surgery. The HFS diet is not recommended during pre- and post-operative therapy, while the control diet (CD) is more favorable in respect of oxidant–antioxidant status following bariatric surgery. The results agree with those presented in a previous study [31].

Lipofuscin is an undegradable material, comprising the residues of oxidized lipids and proteins, which accumulate with time, proportionally to ROS formation intensity [41,42]. As the product of unsaturated fatty acid oxidation, its presence may indicate membrane or mitochondria and lysosome damage. Lipofuscin presence is assumed to result from chronic vitamin E deficiency, caused by tocopherol malabsorption in, e.g., celiac disease, short bowel syndrome, and malabsorptive surgeries [43]. In the present study, lipofuscin levels were the lowest in DJOS-operated rats fed the control diet. Perhaps, lipofuscin plasma concentration would increase with time after DJOS surgery, but nevertheless, this hypothesis should be tested in the dynamic type of a study. 

Some studies show supplementation is a strong modulator of the antioxidative–oxidative balance. A longitudinal study by Jungert et al. demonstrated that plasma concentrations of vitamin C and E can be maintained within the accepted reference ranges with a regular diet and an active lifestyle along the trajectory of advanced aging [44]. Jungert et al. found a positive interrelation between plasma concentrations of vitamins C and E in community-dwelling older adults under everyday conditions as they aged. Although the use of supplements, physical activity, body composition, serum cholesterol levels, diseases, and drugs may modify plasma concentrations of either vitamin C or E, their results indicate that the interrelation between the plasma concentrations of both vitamins is largely independent of these factors [44].

The interpretation of the presented findings has some limitations, the main limitation being the small number of animals enrolled in the study. Moreover, the measured selected parameter in the plasma do not present the complete picture of enzymatic and non-enzymatic oxidative stress markers. 

## 5. Conclusions

Cafeteria diet (HFS) used in the presented study negatively affected the positive impact of DJOS surgery. Nevertheless, eight weeks of dietary treatment before and after the surgery did not totally dilapidate the effects of the bariatric treatment, as the enzymatic (SOD and CER) and non-enzymatic antioxidant systems (MDA and LS) were significantly lower in rats that underwent DJOS bariatric surgery when compared to rats subjected to control surgery. The diet change after the surgery (CD/HFS or HFS/CD) strongly influenced SOD activity in DJOS- and SHAM-operated rats, but SOD activity was always lower in DJOS-operated rats. Every dietary protocol used in the study increased CER activity, except for the CD/CD combination. Cafeteria diet consumed before or after either of the surgeries led to decrease in TAC levels. DJOS and no change in diet reduced MDA levels in rats’ plasma. DJOS reduced LS levels, but its beneficial effect was deteriorated by selected dietary protocols. Strong dietary support for the patients after bariatric surgery and possible support of antioxidant systems seems crucial to achieving the best-expected results of bariatric surgery.

## Figures and Tables

**Figure 1 nutrients-14-04097-f001:**
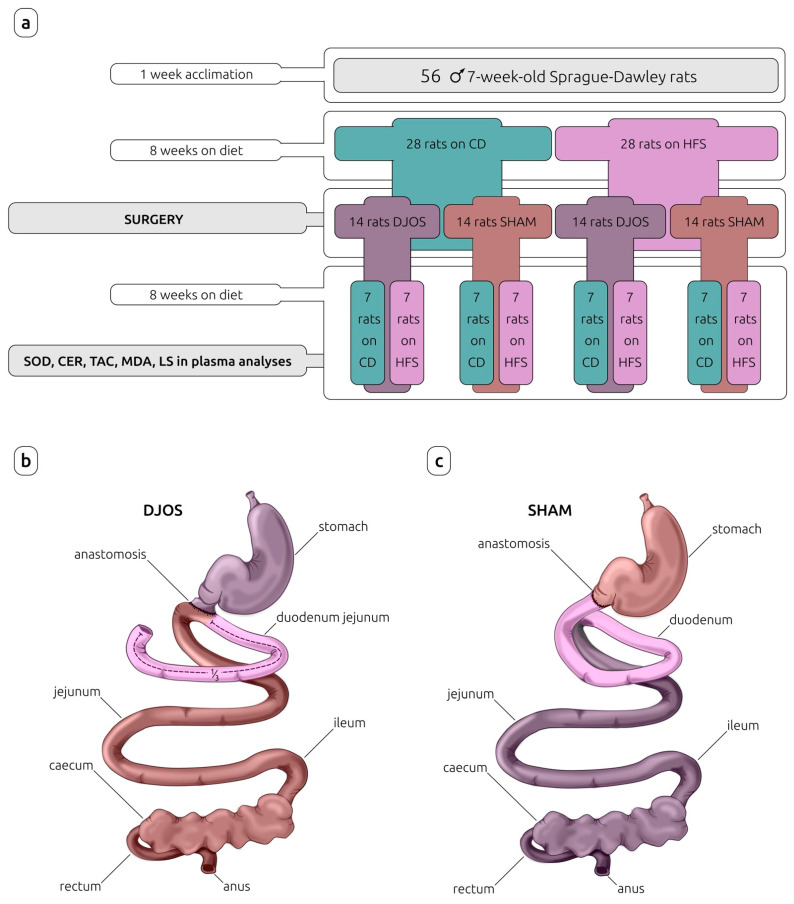
Study design (**a**) and surgeries performed on Sprague-Dawley rats subjected to (**b**) duodenojejunal omega switch (DJOS) and (**c**) control (SHAM) surgery. Legend: CD—control diet, CER—ceruloplasmin activity, HFS—high-fat/high-sugar diet, LS—lipofuscin concentration, MDA—malondialdehyde concentration, SOD—superoxide dismutase activity, TAC—total antioxidant capacity.

**Figure 2 nutrients-14-04097-f002:**
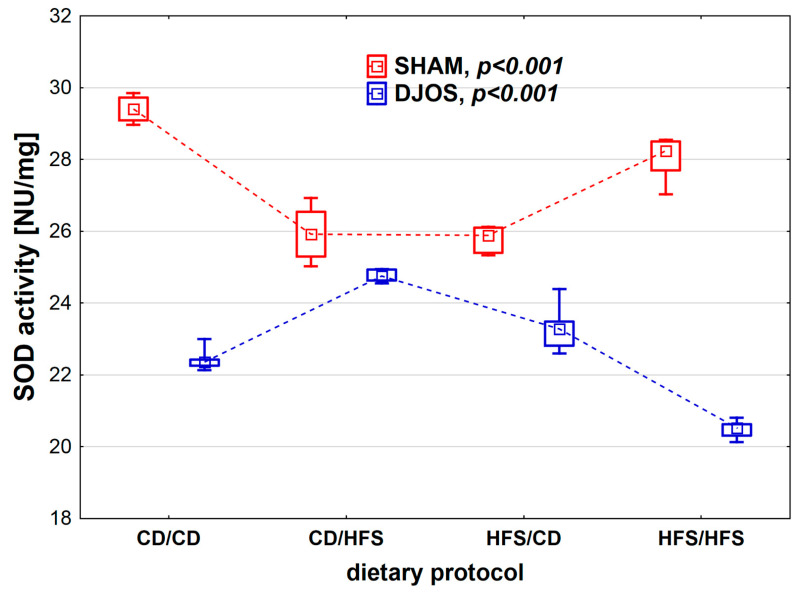
Superoxide dismutase (SOD) activity (NU/mg) in the plasma of Sprague-Dawley rats subjected to duodenojejunal omega switch (DJOS) and control (SHAM) surgery and different dietary protocols at 8 weeks after the surgery. Legend: CD—control diet, HFS—high fat/high sugar diet, NU—nitrate unit. The inner boxes represent the medians, the outer boxes represent interquartile ranges, and whiskers represent min–max range. For the reader’s convenience, the medians are connected with dashed lines.

**Figure 3 nutrients-14-04097-f003:**
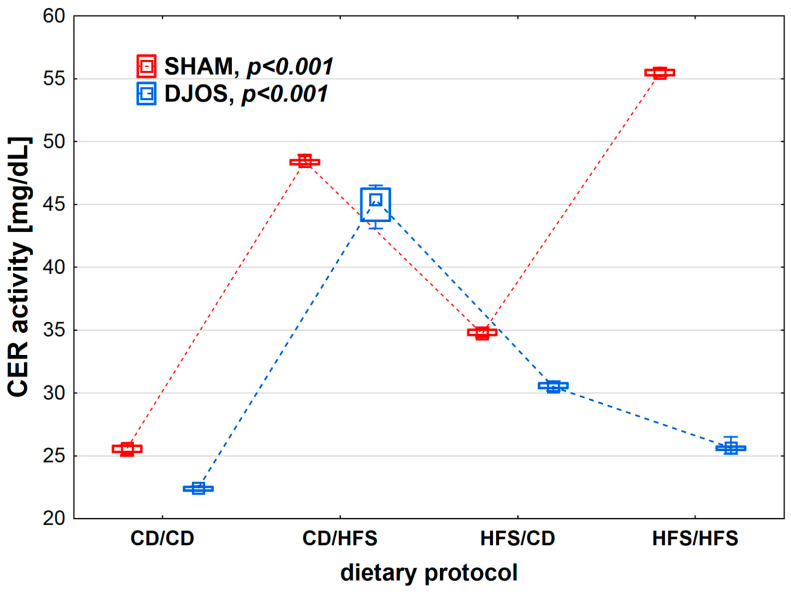
Ceruloplasmin (CER) activity (mg/mL) in the plasma of Sprague-Dawley rats subjected to duodenojejunal omega switch (DJOS) and control (SHAM) surgery and different dietary protocols at 8 weeks after the surgery. Legend: CD—control diet, HFS—high-fat/high-sugar diet. The inner boxes represent the medians, the outer boxes represent interquartile ranges, and whiskers represent min–max range. For the reader’s convenience, the medians are connected with dashed lines.

**Figure 4 nutrients-14-04097-f004:**
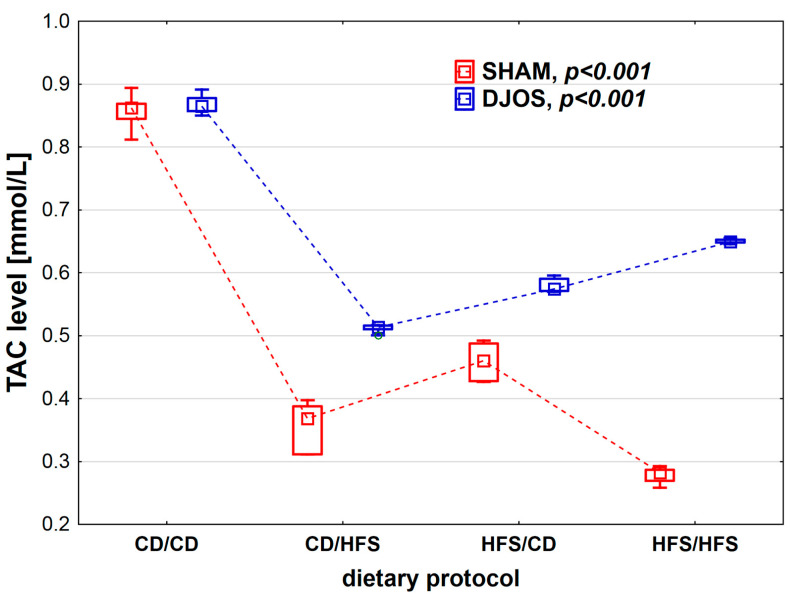
Total antioxidant capacity (TAC) level (mmol/L) in the plasma of Sprague-Dawley rats subjected to duodenojejunal omega switch (DJOS) and control (SHAM) surgery and different dietary protocols at 8 weeks after the surgery. Legend: CD—control diet, HFS—high-fat/high-sugar diet. The inner boxes represent the medians, the outer boxes represent interquartile ranges, and whiskers represent min–max range. For the reader’s convenience, the medians are connected with dashed lines.

**Figure 5 nutrients-14-04097-f005:**
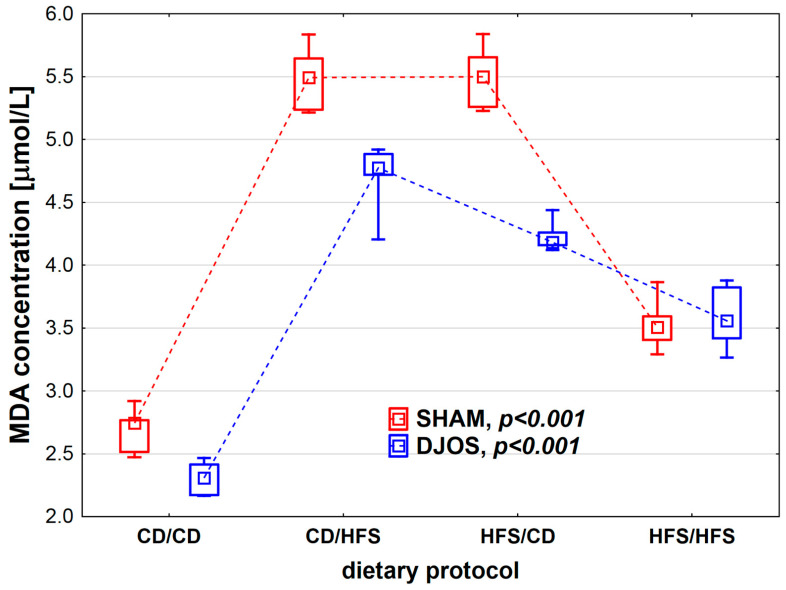
Malondialdehyde (MDA) concentration (µmol/L) in the plasma of Sprague-Dawley rats subjected to duodenojejunal omega switch (DJOS) and control (SHAM) surgery and different dietary protocols at 8 weeks after the surgery. Legend: CD—control diet, HFS—high-fat/high-sugar diet. The inner boxes represent the medians, the outer boxes represent interquartile ranges, and whiskers represent min–max range. For the reader’s convenience, the medians are connected with dashed lines.

**Figure 6 nutrients-14-04097-f006:**
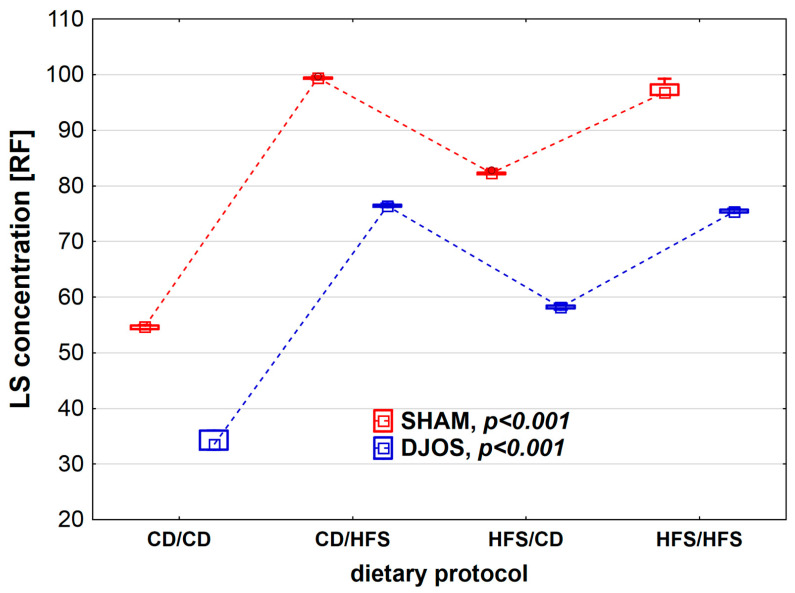
Lipofuscin (LS) concentration (RF) in the plasma of Sprague-Dawley rats subjected to duodenojejunal omega switch (DJOS) and control (SHAM) surgery and different dietary protocols at 8 weeks after the surgery. Legend: CD—control diet, HFS—high-fat/high-sugar diet, RF—relative lipid extract fluorescence. The inner boxes represent the medians, the outer boxes represent interquartile ranges, and whiskers represent min–max range. For the reader’s convenience, the medians are connected with dashed lines.

**Table 1 nutrients-14-04097-t001:** Characteristics of control (CD) and high-fat/high-sugar (HFS) diets fed to Sprague-Dawley rats included in to the study.

Fodder Characteristics	CD Diet	HFS Diet
Sugar/Dextrines (%)	-	29.4
Starch (%)	29.0	8.6
Protein (%)	24.0	22.5
Fat (%)	4.9	23.1
Crude ash (%)	7.0	5.9
Crude fiber (%)	4.7	5.7
Energy value (MJ/kg)	16.5	22.1
Fodder name	3800 Breeding Standard	ssniff^®^ EF R/M acc. D12451 (II) mod.
Product origin	Provimi Kliba AG, Kaiseraugst, Switzerland	ssniff Spezialdiäten GmbH, Soest, Germany

## Data Availability

The data will be available after contact with the corresponding author.

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
