# Peer review of "Duodenojejunal Omega Switch Surgery Reduces Oxidative Stress Induced by Cafeteria Diet in Sprague-Dawley Rats"

_nutrients, 2022, doi:10.3390/nu14194097_

Round 1
Reviewer 1 Report
Title: Duodenojejunal Omega Switch Surgery Reduces Oxidative Stress Induced by Cafeteria Diet in Sprague-Dawley Rats
Although study looks interesting there are some issues with this manuscript.
1. English language throughout the manuscript needs to be improved.
2. Introduction:
-I would advise beginning the introduction with a brief description of diabetes, corelating with obesity, insulin resistance and type 2 diabetes. Also answer the question rising from reader behind the role insulin in fat disposition.
- The authors should clarify the novelty of this article in the ‘Introduction’ and ‘Conclusion’ section.
3. Materials and Methods
-Authors need to include the manufacturer, supplier, country, and city for the reagents/instruments used for these studies.
4. Results:
- These sections need to be written in the brief rather than descriptive.
-Please include the significant symbol (*) within the figures.
5. Discussion:
-How is this article more informative than the previously published ones? Justify it.
6. Conclusion:
- This section does not sound good; therefore, authors have to rewrite this section again by summarizing the outcomes including their future prospective and limitations of their findings.
Author Response
- English language throughout the manuscript needs to be improved.
Answer:
The manuscript has been corrected by an independent English editing service.
- Introduction:
-I would advise beginning the introduction with a brief description of diabetes, corelating with obesity, insulin resistance and type 2 diabetes. Also answer the question rising from reader behind the role insulin in fat disposition.
Answer:
We have improved the Introduction section with the suggested information. However, we focused more on relationship between obesity and oxidative stress effects accompanying it.
- The authors should clarify the novelty of this article in the ‘Introduction’ and ‘Conclusion’ section
Answer:
The Introduction and Conclusions sections have been extended with the requested information.
- Materials and Methods
-Authors need to include the manufacturer, supplier, country, and city for the reagents/instruments used for these studies.
Answer:
The manuscript has been improved in the context of this request – we have provided the complete information for all reagents and instruments used in the experiment except for the oxidative stress markers analyses, since we do not describe in details the assay protocols.
- Results:
- These sections need to be written in the brief rather than descriptive.
Answer:
Due to the complexity of the experimental setup, the tested variables (2 operations and 4 dietary regimens for), and interactions between the variables it is not possible to describe the results in brief. We decided to leave the Results section as it is, so readers of all backgrounds are able to easily understand it.
-Please include the significant symbol (*) within the figures.
Answer:
Figures have been improved. However, due to the reasons described above it is impossible to include all significant differences in the Figures. The Supplementary Material contains 2 Tables with the results of all analyses.
- Discussion:
-How is this article more informative than the previously published ones? Justify it.
Answer:
The submitted article presents the results of the project evaluating the effects of cafeteria diet on the bariatric surgery outcome in animals with diet-induced obesity.
Our observations from the previous experiments conducted onca high fat (HF) diet led us to conclusions that diet is a stronger factor for changes in metabolic processes that surgical interventions. The main aim of currently presented cafeteria diet (high fat/high sugar, HFS) experiment was to assess whether HFS diet has similar effects as HF diet.
- Conclusion:
- This section does not sound good; therefore, authors have to rewrite this section again by summarizing the outcomes including their future prospective and limitations of their findings.
Answer:
The Conclusions section has been improved.

Reviewer 2 Report
1. My biggest worry is the significance of these results. The authors must explain why they chose to conduct the measurements in the animals' plasma. Could this be used to support a pharmaceutical therapy, for example?
2. The activity of a single antioxidant enzyme, superoxide dismutase, was evaluated by the authors. Other antioxidant enzymes, including as catalase and glutathione reductase, are thought to be key oxidative stress defenders. Wouldn't it be interesting to also show the activity of these enzymes? Is there something specific that prompted the selection of superoxide dismutase?
3. Specific antioxidant plasma dosages, such as vitamin C and vitamin E, may be beneficial in understanding the findings given. Is there a bibliographic reference available that might be utilized in this case?
4. Is it feasible to measure GSH and GSSG levels in plasma? If this is the case, such analyses would be extremely valuable, given that such compounds are directly connected to the oxidative stress profile of certain samples.
5. Malondialdehyde levels are proportional to the rate of lipid peroxidation. This was extremely nicely argued by the authors. However, no protein damage signals, such as carbonylated proteins, were shown. This investigation would also be relevant given that carbonylation might damage the functioning of enzymes like superoxide dismutase.
6. Unless I'm wrong, the statistical analyses aren't completely visible in the figures. This would undoubtedly make reading and interpreting the results easier.
Author Response
Reply to Review Reports
We would like to thank the reviewers for their valuable comments on our manuscript. Each comment has been
carefully considered and responded to. All changes are highlighted in blue in the manuscript.
The responses to individual comments are given below.
The authors
- My biggest worry is the significance of these results. The authors must explain why they chose to conduct the measurements in the animals' plasma. Could this be used to support a pharmaceutical therapy, for example?
Answer:
Obesity is associated with an increase in oxidative stress, which is defined as an increased load of free radicals composed of reactive oxygen and nitrogen species generated during cellular metabolism. These chemically reactive molecules can damage cell proteins, membranes, and DNA. Oxidative stress is suspected to be involved in the pathogenesis of obesity-associated insulin resistance. Systemic enzymatic and non-enzymatic antioxidative systems give the fastest and simplest answer about the condition of antioxidative systems, which, if necessary, can be supported or moderated by pharmacological supplementation and other types of therapy.
- The activity of a single antioxidant enzyme, superoxide dismutase, was evaluated by the authors. Other antioxidant enzymes, including as catalase and glutathione reductase, are thought to be key oxidative stress defenders. Wouldn't it be interesting to also show the activity of these enzymes? Is there something specific that prompted the selection of superoxide dismutase?
Answer:
We are thankful for this comment. Not all markers can be assayed in serum or plasma. In the case of our study, catalase, glutathione reductase, and glutathione peroxidase could not have been assessed in the plasma due to the small amount of collected plasma. The small amount of collected plasma is related to a small amount of blood that can be collected from the rats during one blood collection without influencing the animal’s well-being. Please note that the end result of the experiment was that the animals stayed alive. Nevertheless, we are working on analysis in different tissues to help evaluate more enzymatic and non-enzymatic antioxidative systems in cafeteria diet conditions.
- Specific antioxidant plasma dosages, such as vitamin C and vitamin E, may be beneficial in understanding the findings given. Is there a bibliographic reference available that might be utilized in this case?
Answer: Thank you for that suggestion. We have added relevant information in the revised version of the manuscript.
- Is it feasible to measure GSH and GSSG levels in plasma? If this is the case, such analyses would be extremely valuable, given that such compounds are directly connected to the oxidative stress profile of certain samples.
Answer:
Thank you very much for this suggestion. We are planning to assess GSH and GSSG in our future analyses.
- Malondialdehyde levels are proportional to the rate of lipid peroxidation. This was extremely nicely argued by the authors. However, no protein damage signals, such as carbonylated proteins, were shown. This investigation would also be relevant given that carbonylation might damage the functioning of enzymes like superoxide dismutase.
Answer:
Thank you very much for this suggestion. We will definitely include this type of analysis in our future studies.
- Unless I'm wrong, the statistical analyses aren't completely visible in the figures. This would undoubtedly make reading and interpreting the results easier.
Answer:
Due to the complexity of the experimental setup, the tested variables (2 operations and 4 dietary regimens for), and interactions between the variables it was not possible to present the statistical analysis results in the Figures. Instead, we decided to describe the results extensively, so the readers of all backgrounds are able to easily understand it. However, we decided to present the results of comparisons for dietary regimens within the subgroups of rats subjected to two different types of surgery. The Supplementary Material contains a Table with the results of all analyses.
